# Retinal Microvascular Signs in Pre- and Early-Stage Diabetic Retinopathy Detected Using Wide-Field Swept-Source Optical Coherence Tomographic Angiography

**DOI:** 10.3390/jcm11154332

**Published:** 2022-07-26

**Authors:** Fabao Xu, Zhiwen Li, Yang Gao, Xueying Yang, Ziyuan Huang, Zhiwei Li, Rui Zhang, Shaopeng Wang, Xinghong Guo, Xinguo Hou, Xiaolin Ning, Jianqiao Li

**Affiliations:** 1Department of Ophthalmology, Qilu Hospital, Shandong University, Jinan 250012, China; xufabao09591@qiluhospital.com (F.X.); 202136249@mail.sdu.edu.cn (Z.L.); 202136253@mail.sdu.edu.cn (X.Y.); rui.zhang99@sdu.edu.cn (R.Z.); 2Shandong Key Laboratory: Magnetic Field-Free Medicine & Functional Imaging, Jinan 250000, China; yanggao@buaa.edu.cn (Y.G.); huangziyuan@buaa.edu.cn (Z.H.); ningxiaolin@buaa.edu.cn (X.N.); 3Magnetic Field-Free Medicine & Functional Imaging, Research Institute of Shandong University, Jinan 250000, China; 4School of Physics, Beihang University, Beijing 100191, China; 5Department of Ophthalmology, Jinan Aier Eye Hospital, Jinan 250000, China; lizhiwei2@aierchina.com; 6Department of Ophthalmology, Zibo Central Hospital, Binzhou Medical University, Zibo 250012, China; wangshaopeng@medmail.com.cn; 7Department of Endocrinology, Qilu Hospital, Shandong University, Jinan 255000, China; guoxinghong09541@qiluhospital.com (X.G.); houxinguo@sdu.edu.cn (X.H.)

**Keywords:** diabetes mellitus, diabetic retinopathy, wide-field optical coherence tomographic angiography, retinal vascular changes

## Abstract

**Purpose** Using a wide-field, high-resolution swept-source optical coherence tomographic angiography (OCTA), this study investigated microvascular abnormalities in patients with pre- and early-stage diabetic retinopathy. **Methods** 38 eyes of 20 people with diabetes mellitus (DM) type 2 without diabetic retinopathy (DR) and 39 eyes of 21 people with DR were enrolled in this observational and cross-sectional cohort study, and a refractive error-matched group consisting of 42 eyes of 21 non-diabetic subjects of similar age were set as the control. Each participant underwent a wide-field swept-source OCTA. On OCTA scans (1.2 cm × 1.2 cm), the mean central macular thickness (CMT), the vessel density of the inner retina, superficial capillary plexus (SCP), and deep capillary plexus (DCP) were independently measured in the whole area (1.2 cm diameter) via concentric rings with varying radii (0–0.3, 0.3–0.6, 0.6–0.9, and 0.9–1.2 cm). **Results** Patients whose eyes had pre-and early-stage DR showed significantly decreased vessel density in the inner retina, SCP, DCP and CMT (early-stage DR) compared with the control. In addition, compared with the average values upon wide-field OCTA, the decreases were even more pronounced for concentric rings with a radius of 0.9–1.2 cm in terms of the inner retina, SCP, DCP and CMT. **Conclusions** Widefield OCTA allows for a more thorough assessment of retinal changes in patients with pre- and early-stage DR.; retinal microvascular abnormalities were observed in both groups. In addition, the decreases in retinal vessel density were more significant in the peripheral concentric ring with a radius of 0.9–1.2 cm. The application of novel and wide-field OCTA could potentially help to detect earlier diabetic microvascular abnormalities.

## 1. Introduction

The number of patients with diabetes mellitus (DM) has quadrupled in the last three decades, and DM is the ninth major cause of death worldwide [1,2,3]. Current evidence suggests that about 1 in 11 adults worldwide now have DM, 90% of whom have type 2 DM (T2DM). Diabetes affects around 366 million people globally, and DR is one of the severe complications of the disease [4,5,6,7]. DR is the leading cause of blindness in the working-age population and is characterized by microaneurysms, hemorrhages, and hard exudates. As a result, early detection and accurate staging of DR are critical to avoid vision-threatening complications [8,9,10]. The progression of pathophysiologic changes of DR may result in diabetic macular edema (DME) and proliferative diabetic retinopathy (PDR), which accounts predominantly for DR-related vision loss [11,12,13,14]. Ophthalmoscopy and color fundus photography are well-established as the golden reference for diagnosing and staging DR. Nonetheless, microvascular damage, including pericyte loss, capillary leakage and nonperfusion, is known to happen before findings of retinopathy become apparent on clinical examination or fundus imaging [1,2,15,16] A growing body of evidence from recently published studies revealed that ophthalmoscopy could detect microvascular changes that manifest prior to the classical features of DR [17,18,19,20,21].

Although fundus fluorescein angiography (FFA) remains the mainstay diagnostic tool for retinal vascular pathology evaluation and the second pillar for diagnosing DR, it is not time efficient, not quantitative, and can bring about dye-related adverse events, and is not practical as a regular follow-up tool [15,18,22]. Moreover, FFA could only detect two-dimensional vascular information, and the depth of investigation does not include deep capillaries [22,23]. OCTA, on the other hand, is a new imaging modality that can be combined with en-face OCT-based techniques to allow visualization of the retina layer without the need for exogenous dyes [12,14,18,24]. OCTA can be used to measure the dimensions of the retinal capillary networks in a non-invasive manner. Several studies that used OCTA revealed that microvascular changes could manifest before the characteristic features of DR are detectable using ophthalmoscopy [8,25]. However, relatively few investigations have focused on a comprehensive quantitative assessment of both pre-and early-stage retinal vascular abnormalities in diabetic patients.

Moreover, the rapid development of OCT and OCTA technologies has provided us with higher-resolution images as well as quantitative tools to investigate vascular changes in the retina more precisely [8,12,25]. Previous OCTA findings in DM and DR patients were limited to specific areas and were primarily macular scans of 0.3 cm × 0.3 cm and 0.6 cm × 0.6 cm. To date, only a handful of reports have investigated vascular disturbances and structural alternations using wide-field imaging with a scan area of 1.2 cm × 1.2 cm (Examples of OCTA images with different ranges are shown in Figure 1) [8,18,26]. In addition, with the development of technology, the miniaturization of equipment and the improvement of information acquisition speed, OCTA, as a more sensitive screening method, will have broad prospects in the field of telemedicine.Therefore, the purpose of this study was to comprehensively evaluate the retinal changes in pre-and early-stage DR using a bigger field of view. Wide-field OCTA with a 1.2 cm × 1.2 cm scanning protocol was used to reach the fundus’ posterior pole.

## 2. Method

This cross-sectional observational study included T2DM patients who visited the Department of Ophthalmology and Department of endocrinology, Qilu Hospital, Cheeloo College of Medicine, Shandong University, and Department of Ophthalmology, Aier Eye Hospital, Jinan, China, from 1 April 2021, to 1 March 2022. Written informed consents (Version 1.0, 20210605001) were requested from all participating subjects. The study was conducted per the Declaration of Helsinki and approved by the Medical Ethics Committee of Qilu Hospital and Aier Eye Hospital (Ethical code: KYLL-202111-024-1 and JALL2022-1).

### 2.1. Inclusion and Exclusion Criteria

Subjects with a random plasma glucose concentration of ≥200 mg/dL, a fasting plasma glucose concentration of ≥126 mg/dL or a 2-h plasma glucose concentration of ≥200 mg/dL after drinking 75-g oral glucose were diagnosed with Type 2 Diabetes. The following criteria were used to determine inclusion: (1) Patients with a definite diagnosis of T2DM, (2) Patients aged over 18, (3) Patients with no DR (NDR) or early-stage DR, (4) Patients with best-corrected visual acuity (BCVA) ≥ 80 letters or better in each eye and (5) OCTA and blood tests were performed between 1 April 2021, to 1 March 2022. The following were the exclusion criteria: (1) clinically significant DME, (2) previous diagnosis of glaucoma and presence of myopia and chorioretinal atrophy and other ocular diseases which might influence the chorioretinal capillaries in the posterior pole, (3) ocular hypertension, (4) use of medications that may have an impact on the vasculature, (5) history of other retinal diseases and vitreoretinal surgery, (6) severe non-proliferative DR, (NPDR) or proliferative DR (PDR) and (7) patients with insufficient medical data.

The participants were assigned to two cohorts, the pre-DR group and the early-stage DR group, based on the pathological features of DR as revealed by fundus photography. According to the severity scale standard approved by the Early Treatment Diabetic Retinopathy Study (ETDRS), pre-DR was defined as the absence of all DR features, and early-stage DR was defined as the presence of a microaneurysm [27,28]. Moreover, an age- and a refractive error-matched control group of 42 eyes from 21 non-diabetic subjects were enrolled.

### 2.2. Ophthalmic Examinations and Blood Test

Each patient and healthy subject underwent a detailed ophthalmic examination and blood test; the ophthalmic examinations included BCVA measurement, intraocular pressure by non-contact tonometer, slit lamp-based biomicroscopy of the anterior segment, and dilated fundus biomicroscopy after full dilation of the pupil. OCTA images were collected using a commercial wide-field OCTA instrument from SVision Imaging with a 1050 nm wavelength and a 200,000 A-scans/s scanning rate. The device had a full width at half maximum axial resolution of around 5 μm in tissue and an estimated lateral resolution of around 15 μm at the retinal surface. OCTA was performed using raster scanning (512 (horizontal) × 512 (vertical)) and B-scans (1024 × 1024), which covered an area of 1.2 cm × 12 cm centered on the fovea. The built-in software determined the OCTA’s image quality from Q1 to Q10. Eyes images that were of low grade (quality index < 6) due to poor transparency and fixation were not included. To avoid possible diurnal variation in the retinal capillaries, all images were taken between 8:00 AM and 11:00 AM by experienced technicians blinded to the subjects’ clinical status. Blood tests included routine blood tests, fasting blood glucose tests, hemoglobin a1c, blood lipids, etc.

## 3. OCTA Image Analysis

For the quantitative analysis of OCTA scans, retinal parameters, which include central macular thickness (CMT), vessel density of inner retina, superficial capillary plexus (SCP), and deep capillary plexus (DCP), were independently quantified centrally and peripherally, accompanied the central fovea area (0.3 cm in diameter) and the perifoveal concentric ring (0.3–0.6 cm), pararetinal concentric ring (0.6–0.9 cm), and periretinal concentric ring (0.9–1.2 cm) using software version 1.32.9 (VG200; SVision Imaging, Ltd., Luoyang, China) (Figure 2). Besides, layer segmentation and quantification analyses of vessel density of the inner retina, SCP, and DCP were also conducted using the same software. Manual manipulation of segmentation was also done when needed to ensure accuracy. CMT was defined as the outer surface of the line formed by the RPE to the outer surface of the retinal nerve fiber layer in the 1.2 cm × 1.2 cm scans area (Figure 3). In addition, vessel density was measured in terms of the percentage of pixels with a flow signal higher than the threshold (%) [9,18]. A panel of two licensed Chinese retinal specialists (Fabao Xu and Zhiwen Li) extracted the OCTA measurement data using the built-in VG200 SVision Imaging software.

### Outcome Measures

The study’s primary outcome measures were the alterations in vessel density of the inner retina measured by OCTA. Moreover, the secondary outcomes were the differences in vessel density of SCP, DCP, and CMT and the comparison of capillaries changes in different retinal zones (Circle_x-y_—Average density/CMT).

## 4. Statistical Analysis

All statistical analysis in this study was conducted using SPSS software (version 19.0 SPSS, Inc, Chicago, IL, USA). For normality testing, the Kolmogorov-Smirnov test was used. Non-parametric variables are represented by the median and interquartile range, while normal variables are represented by mean values and standard deviation (SD) (IQR). A paired-sample *t*-test was used to compare the number of patients enrolled, mean ages, BCVA, durations, and other demographic and baseline characteristics between DM patients and healthy controls. Binocular data were included in all groups of this study. A generalized estimating equation (GEE) statistical method was used to compare normally distributed quantitative variables. A two-tailed *p*-value < 0.05 was deemed statistically significant.

## 5. Results

### 5.1. Baseline Demographic Data

A total of 77 eyes from 41 DM patients (38 eyes from 20 patients in the pre-DR group and 39 eyes from 21 subjects in early-stage DR group) and 42 eyes from 21 age-and refractive error-matched control subjects were enrolled in this study. The demographic data of three groups are shown in Table 1. Five diabetic subjects’ eyes were excluded due to clinical signs of NPDR on fundography and optical media opacity caused by cataracts. The study and control groups did not differ significantly in the number of subjects (*p* = 0.953 in the pre-DR group, *p* = 1 in the early-stage DR group, respectively), age (54.21 ± 9.18 years vs. 56.56 ± 10.87 years, *p* = 0.809; 54.21 ± 9.18 years vs. 57.20 ± 12.07 years, *p* = 0.743), BCVA (82.24 ± 2.31 letters vs. 81.34 ± 1.98 letters, *p* = 0.962; 82.24 ± 2.31 letters vs. 82.41 ± 1.93 letters, *p* = 0.984), CMT (198.34 ± 11.36 µm vs. 202.46 ± 12.17 µm, *p* = 0.904; 198.34 ± 11.36 μm vs. 192.37 ± 10.95 µm, *p* = 0.837), ChT (267.36 ± 29.55 µm vs. 261.30 ± 30.11 µm, *p* = 0.914; 267.36 ± 29.55 µm vs. 256.07 ± 33.47 µm, *p* = 0.722) and other blood test values (Table 1).

### 5.2. Quantitative Evaluation of the Inner Retina

The study and control groups exhibited no significant differences in average vessel density of the inner retinal vascular layer. However, the study group was associated with significantly lower vessel density during subgroup analysis based on the central fovea area (diameter of 3 mm) and the perifoveal concentric ring (3–6, 6–9 and 9–12 mm). The central fovea area with a diameter of 0–3 mm (*p* = 0.016) and concentric ring of 9–12 mm (*p* = 0.008) was associated with a significantly decreased vessel density between the control and pre-DR groups. The central fovea area (*p* = 0.002) and all concentric rings (*p* = 0.007, *p* = 0.015, *p* < 0.001, respectively) showed significant decreases in vessel density between the control and early-stage DR groups (Table 2 and Figure 4).

### 5.3. Quantitative Evaluation of SCP

The early-stage DR group showed a significantly lower average SCP than the control group (*p* = 0.041). During the subgroup analysis of different areas, the central fovea area with a diameter of 0–3 mm (*p* = 0.021) and concentric ring at 9–12 mm (*p* = 0.002) exhibited significantly decreased vessel density between the normal control and pre-DR group. The central fovea area (*p* < 0.001) and all concentric rings (*p* < 0.001, *p* = 0.037, *p* < 0.001, respectively) showed significantly decreased vessel density between the control and early-stage DR groups (Table 2 and Figure 4).

### 5.4. Quantitative Evaluation of DCP

The early-stage DR group was linked to a markedly lower average DCP in comparison to the control group (*p* < 0.001). During the subgroup analysis of different areas, the central fovea area with a diameter of 0–3 mm (*p* = 0.033) and concentric ring at 3–6 mm (*p* < 0.001) showed significantly decreased vessel density between the control and pre-DR groups; the central fovea area (*p* < 0.001) and all concentric rings (*p* < 0.001, respectively) showed significantly decreased vessel density between the control and early-stage DR groups (Table 2 and Figure 4).

### 5.5. Quantitative Evaluation of CMT

During subgroup analysis, the retinal thickness decreased significantly only in the concentric ring of 9–12 mm in the early-stage DR group compared to the control group (*p* = 0.024) (Table 2 and Figure 4 in detail).

### 5.6. Qualitative Evaluation of the Concentric Ring Area

To compare the influence of the retinal structure in different areas away from the central fovea, we calculated the differences in vessel density between the concentric rings and average values of different retinal layers such as inner retina, SCP and DCP and the differences in retinal thickness between the concentric rings and average retinal thickness. The 9–12 mm concentric ring significantly decreased when comparing the inner retina and SCP between the pre-DR and control groups. Moreover, the peripheral retinal ring of 9–12 mm showed a significant decrease in all parameters (inner retina, SCP, DCP and CMT) between the early-stage DR and control groups (Table 3 and Figure 5).

## 6. Discussion

This study conducted a cross-sectional analysis of OCTA-derived parameters in T2DM patients with pre- or early-stage DR. Importantly, we found that micro-retina vascular changes can be detected before the characteristic findings of DR can be seen upon ophthalmoscopy or fundography. The vessel density of the inner retina, SCP and DCP in patients with pre-and early-stage DR were significantly thinner in contrast to the control group. Moreover, the vessel density of the peripheral retinal ring of 9–12 mm was significantly decreased compared to the concentric rings near the fovea.

It is widely acknowledged that OCTA scanning is increasingly fast with a broader scope. With significant inroads achieved in recent years, different OCTA devices have been developed to visualize retinal and subretinal vascular abnormities in DM patients [26,29,30,31]. The scan pattern of 12 mm × 12 mm is the maximum range currently available for a single scanning of all OCTA devices [29,30,31,32]. Overall, the present study findings were consistent with the literature. Indeed, DM patients with no or mild DR signs show lower retinal vessel density on OCTA in comparison to non-diabetics. In 2017, Simonett and colleagues reported a remarkable decrease in parafoveal vessel density in the DCP of T1DM patients without DR in contrast to non-diabetic controls. In that study, the range of the OCTA device acquisition pattern adopted was only 3 mm × 3 mm [11], which was in agreement with previous reports that investigated DR patients. Additionally, Durbin et al. reported in a prospective cross-sectional study on 50 eyes obtained from 26 diabetic participants and 50 healthy eyes obtained from 25 non-diabetic participants that the retinal vessel density of the SCP showed the highest area under the receiver operating characteristic (ROC) curve when differencing between DR and non-DR groups, followed by the foveal avascular zone (FAZ) area and finally the vessel density of the DCP. These findings indicated that SCP is the optimal parameter for differentiating healthy eyes from eyes with DR [33]. OCTA imaging was performed using a pattern of 3 mm × 3 mm (Carl Zeiss Meditec, Inc., Dublin, CA, USA). However, it should be borne in mind that most previous studies used OCTA devices with a small scanning range. Importantly, the present study provided a more comprehensive assessment with a wide-field OCTA of 12 mm × 12 mm.

We discovered the peripheral features of pre- and early-stage DR eyes using a 1.2 cm × 1.2 cm protocol to scan a wide-field image of the posterior pole. To compare variations in retinal vessel density at different distances away from the central fovea in the 12 mm × 12 mm scanning range, we divided the retinal capillaries into four concentric rings with different diameters (Figure 2) and calculated the differences in vessel density/CMT between the concentric rings and average values of different retinal layers. The results suggested that the 9–12 mm concentric ring decreased most in the early-stage DR group. This phenomenon suggests that the influence of DM on retinal microvascular existed before the appearance of DR and that peripheral retinal involvement may be more evident than that near the central fovea.

Taken together, wide-field OCTA is more appropriate for early screening and monitoring of microvascular abnormalities in diabetic patients. This cutting-edge technology allows a thorough examination of structural and vascular alterations in the central and peripheral retinal areas, expanding our understanding of pre- and early-stage DR pathophysiology.

Interestingly, we found that the peripheral retinal thickness in the early-stage DR group was significantly lower than in the control group. It is well-established that the CMT of early-stage DR is not significantly different from normal subjects [8,9,18]; our results indicate that the peripheral retina may be a more sensitive area, prone to retinal atrophy under the influence of DM. However, the difference may be due to the limited range of OCT scans used in previous studies.

Nevertheless, our study has several limitations. Firstly, the study’s cross-sectional nature restricted the dynamic assessment of retinal alterations to a certain extent with disease progression. Additional longitudinal prospective studies on wide-field OCTA and experimental studies are required to substantiate the current. Besides, this study only enrolled a small sample size of young patients. More research is required to apply our findings to people of all ages. Moreover, we focused on retinal vascular changes in patients with pre-and early-stage DR, while patients with NPDR and PDR were not analyzed, warranting further research to understand the long-term alteration in retinal characteristics with the progression of DR.

In conclusion, we provide a hitherto undocumented report of wide-field OCTA to evaluate retinal microvascular changes in pre-and early-stage DR patients with a 12 cm × 12 cm field of view. The findings revealed that retinal vessel density significantly decreased in both groups, and the decrease was more pronounced in the peripheral concentric ring of 9–12 mm. This study adds to the body of knowledge and offers new insights into pre- and early-stage DR pathophysiology, demonstrating that wide-field OCTA can be a promising and effective modality for clinically assessing retinal abnormalities.

## Figures and Tables

**Figure 1 jcm-11-04332-f001:**
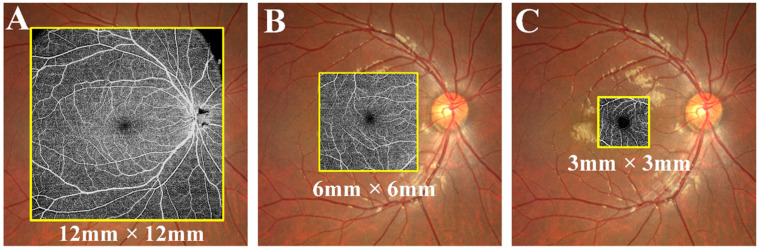
Examples of OCTA images with different ranges are shown in Figure 1. (**A**) Wide-field OCTA image of 12 mm × 12 mm; (**B**) OCTA image of 6 mm × 6 mm; (**C**) OCTA image of 3 mm × 3 mm. OCTA, optical coherence tomographic angiography.

**Figure 2 jcm-11-04332-f002:**
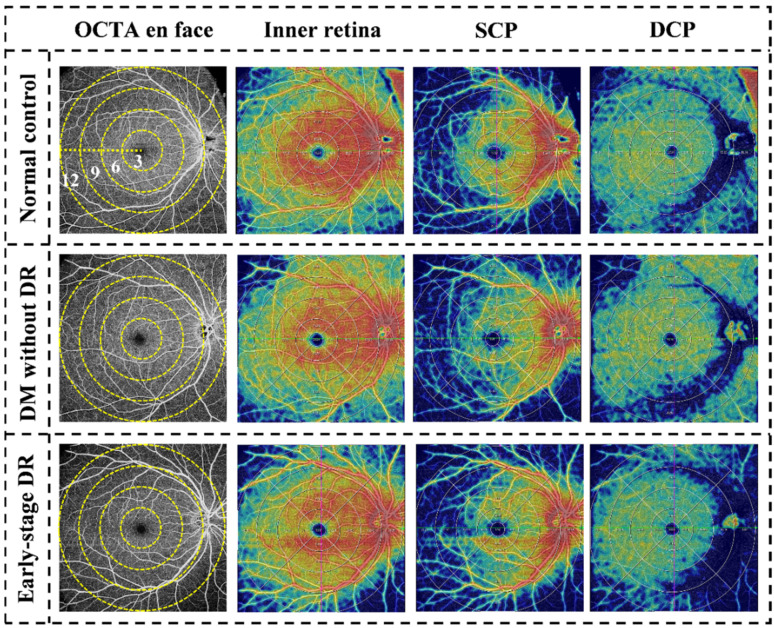
Examples of quantitative measurement using Wide-field OCTA. The left column represents en face images (12 mm × 12 mm) of color code for vessel density. The following columns represent en face images of the corresponding layer of the inner retina, SCP, and DCP of color code for perfusion. The vessel density of the inner retina, SCP, and DCP were separately calculated in the whole area (diameter of 12 mm) and concentric rings with different radii (0–3, 3–6, 6–9, and 9–12 mm). OCTA, optical coherence tomographic angiography; SCP, superficial capillary plexus; DCP, deep capillary plexus.

**Figure 3 jcm-11-04332-f003:**
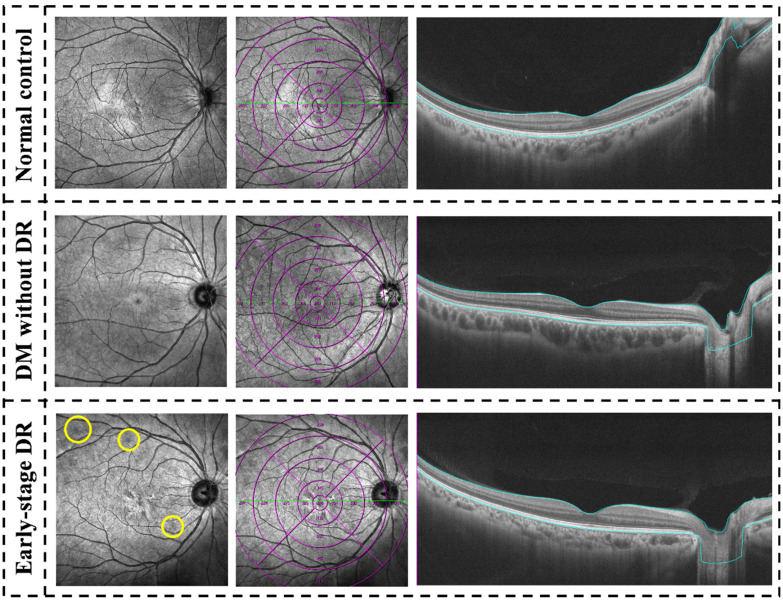
Examples of quantitative measurement of CMT using Wide-field optical coherence tomography in patients with early-stage diabetic retinopathy, diabetes mellitus without ophthalmoscopic signs of diabetic retinopathy and non-diabetic subjects of the control group. The left column was En face infrared photography (12 mm × 12 mm) images, the yellow circle at the lower left showed fundus hemorrhage of early-stage DR. The following columns were quantitative measurements of the whole area (diameter of 12 mm) and in concentric rings with different radii (0–3, 3–6, 6–9, and 9–12 mm) of CMT. CMT, central macular thickness.

**Figure 4 jcm-11-04332-f004:**
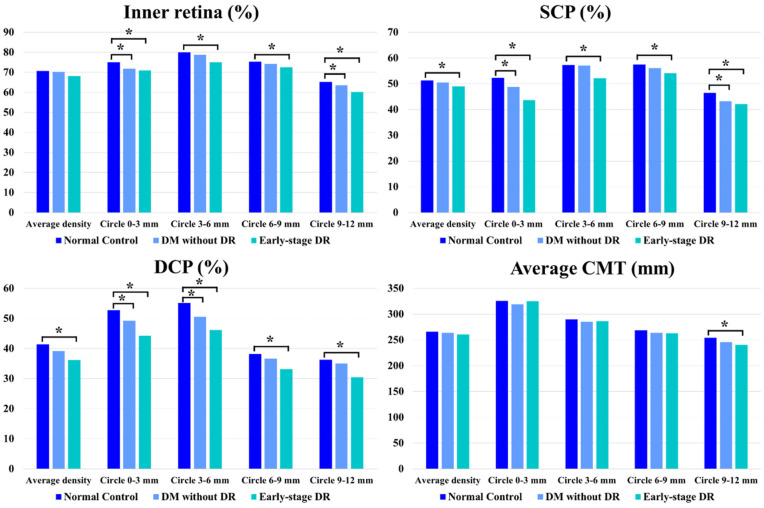
Vessel density and CMT comparisons. Vessel density and CMT comparisons of patients with early-stage diabetic retinopathy, diabetes mellitus without ophthalmoscopic signs of diabetic retinopathy and non-diabetic subjects of the control group. * indicates a statistically significant difference (*p* < 0.05).

**Figure 5 jcm-11-04332-f005:**
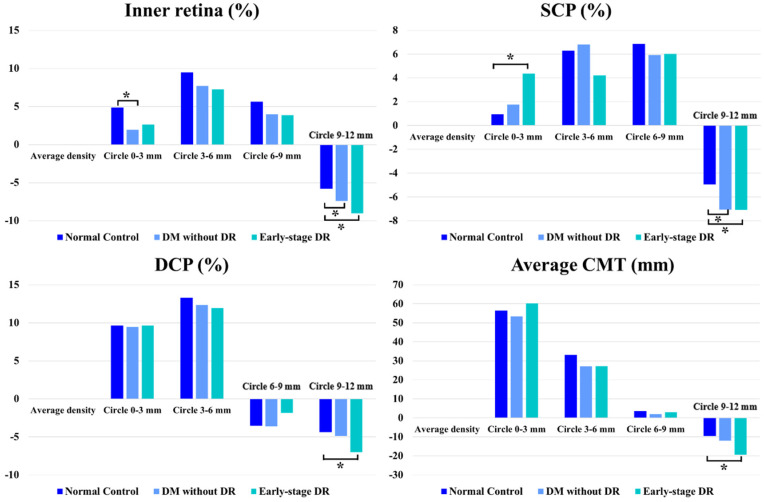
Capillary and CMT changes in different retinal zones. Capillary and CMT changes in different retinal zones (Circle_x-y_ vs. Average density) in patients with early-stage diabetic retinopathy, diabetes mellitus without ophthalmoscopic signs of diabetic retinopathy and non-diabetic subjects of the control group. * indicates a statistically significant difference (*p* < 0.05).

**Table 1 jcm-11-04332-t001:** Demographic characteristics and baseline features of DM patients and healthy controls.

	Normal Control	Pre-DR	*p* Values	Early-Stage DR	*p* Values
Patients (Female)	21 (12)	20 (11)	0.953	21 (11)	1.000
Ages	54.21 ± 9.18	56.56 ± 10.87	0.809	57.20 ± 12.07	0.743
Eyes	42	38	0.825	39	0.864
Duration of DM (months)	N/A	8.14 ± 5.65	N/A	26.02 ± 12.34	N/A
BCVA (ETDRS)	82.24 ± 2.31	81.34 ± 1.98	0.962	82.41 ± 1.93	0.984
GLU	N/A	8.80 ± 7.63	N/A	9.01 ± 8.32	N/A
HbA1c	N/A	6.59 ± 4.06	N/A	8.78 ± 3.21	N/A
CMT (μm)	198.34 ± 11.36	202.46 ± 12.17	0.904	192.37 ± 10.95	0.837
ChT (μm)	267.36 ± 29.55	261.30 ± 30.11	0.914	256.07 ± 33.47	0.722
Image Quality Index	8.39 ± 1.56	8.58 ± 1.49	0.964	8.57 ± 1.63	0.959

Values are shown as means ± SDs. DM, diabetes mellitus; DR, diabetic retinopathy; BCVA, best correct visual acuity; ETDRS, Early Treatment Diabetic Retinopathy Study; GLU, Glucose; HbA1c, Hemoglobin A1C; CMT, central macular thickness; ChT, choroidal thickness. N/A, not applicable. A *p*-value < 0.05 was statistically significant.

**Table 2 jcm-11-04332-t002:** Findings of optical coherence tomographical angiography in subjects with early-stage diabetic retinopathy, DM without ophthalmoscopic signs of DR and non-diabetic control subjects.

Layers	Location	Normal Control	Pre-DR	*p* Values	Early-Stage DR	*p* Values
Inner retina	Average density	70.75 ± 2.28	70.26 ± 3.37	*p* = 0.765	68.09 ± 3.00	*p* = 0.210
	Ring 0–3	75.06 ± 3.01	71.81 ± 6.33	*p* = 0.016 *	70.96 ± 4.65	*p* = 0.002 *
	Ring 3–6	79.99 ± 2.48	78.71 ± 4.57	*p* = 0.542	75.02 ± 4.75	*p* = 0.007 *
	Ring 6–9	75.32 ± 2.85	74.34 ± 3.37	*p* = 0.461	72.50 ± 3.52	*p* = 0.015 *
	Ring 9–12	65.25 ± 3.24	63.44 ± 3.81	*p* = 0.008 *	60.22 ± 3.63	*p* < 0.001 *
SCP	Average density	51.27 ± 3.67	50.48 ± 4.55	*p* = 0.410	48.94 ± 7.87	*p* = 0.041 *
	Ring 0–3	52.28 ± 4.31	48.77 ± 7.57	*p* = 0.021 *	43.67 ± 6.96	*p* < 0.001 *
	Ring 3–6	57.28 ± 4.10	57.08 ± 5.40	*p* = 0.823	52.15 ± 5.81	*p* < 0.001 *
	Ring 6–9	57.38 ± 4.62	56.12 ± 4.40	*p* = 0.548	54.13 ± 9.62	*p* = 0.037 *
	Ring 9–12	46.42 ± 7.49	43.25 ± 6.64	*p* = 0.002 *	42.16 ± 6.94	*p* < 0.001 *
DCP	Average density	41.36 ± 2.64	39.20 ± 4.45	*p* = 0.143	36.23 ± 8.34	*p* < 0.001 *
	Ring 0–3	52.76 ± 4.03	49.22 ± 6.09	*p* = 0.033 *	44.21 ± 8.02	*p* < 0.001 *
	Ring 3–6	55.14 ± 6.52	50.58 ± 6.72	*p* < 0.001 *	46.12 ± 7.63	*p* < 0.001 *
	Ring 6–9	38.19 ± 3.81	36.69 ± 6.27	*p* = 0.260	33.09 ± 6.24	*p* < 0.001 *
	Ring 9–12	36.28 ± 2.78	35.08 ± 5.92	*p* = 0.614	30.43 ± 6.92	*p* < 0.001 *
Average CMT	Average thickness	265.73 ± 8.36	263.74 ± 14.40	*p* = 0.754	261.08 ± 10.12	*p* = 0.716
	Ring 0–3	325.74 ± 14.98	318.99 ± 15.01	*p* = 0.219	325.12 ± 13.68	*p* = 0.890
	Ring 3–6	289.70 ± 10.54	285.39 ± 16.67	*p* = 0.754	286.83 ± 15.62	*p* = 0.773
	Ring 6–9	268.88 ± 10.69	263.70 ± 13.16	*p* = 0.684	263.05 ± 19.98	*p* = 0.681
	Ring 9–12	254.45 ± 9.38	245.54 ± 16.28	*p* = 0.072	240.60 ± 24.39	*p* = 0.024 *

Values are shown as means ± SDs. DM, diabetes mellitus; DR, diabetic retinopathy; SCP, superficial capillary plexus; DCP, deep capillary plexus; CMT, central macular thickness. *p* values represent the comparisons between the Pre-DR and Early-stage DR groups and the normal control group. * indicates a statistically significant difference (*p* < 0.05).

**Table 3 jcm-11-04332-t003:** Comparison of capillaries changes in different retinal zones (Circle_x-y_ vs. Average density) in patients with early-stage diabetic retinopathy, diabetes mellitus without ophthalmoscopic signs of diabetic retinopathy and non-diabetic subjects of the control group.

Layers	Location	Normal Control	Pre-DR	*p* Values	Early-Stage DR	*p* Values
Inner retina	Average density	70.75 ± 2.28	70.26 ± 3.37	N/A	68.09 ± 3.00	N/A
	Circle 0–3	4.89 ± 3.54	1.95 ± 4.36	*p* = 0.021 *	2.62 ± 3.84	*p* = 0.105
	Circle 3–6	9.45 ± 4.32	7.69 ± 4.29	*p* = 0.483	7.25 ± 3.51	*p* = 0.427
	Circle 6–9	5.61 ± 2.84	3.96 ± 2.85	*p* = 0.491	3.87 ± 3.19	*p* = 0.465
	Circle 9–12	−5.81 ± 3.67	−7.41 ± 3.74	*p* = 0.038 *	−9.02 ± 3.42	*p* < 0.001 *
SCP	Average density	51.27 ± 3.67	50.48 ± 4.55	N/A	48.94 ± 7.87	N/A
	Circle 0–3	0.93 ± 3.47	1.75 ± 6.54	*p* = 0.338	4.36 ± 6.81	*p* = 0.013 *
	Circle 3–6	6.28 ± 4.14	6.81 ± 5.83	*p* = 0.851	4.21 ± 6.23	*p* = 0.230
	Circle 6–9	6.85 ± 3.86	5.92 ± 6.02	*p* = 0.705	6.03 ± 7.28	*p* = 0.861
	Circle 9–12	−4.96 ± 6.43	−7.06 ± 7.32	*p* = 0.009 *	−7.09 ± 5.36	*p* = 0.008 *
DCP	Average density	41.36 ± 2.64	39.20 ± 4.45	N/A	36.23 ± 8.34	N/A
	Circle 0–3	9.65 ± 7.02	9.45 ± 6.63	*p* = 0.921	9.65 ± 7.10	*p* = 0.987
	Circle 3–6	13.28 ± 7.74	12.36 ± 7.80	*p* = 0.893	11.96 ± 6.90	*p* = 0.765
	Circle 6–9	−3.53 ± 3.56	−3.62 ± 5.39	*p* = 0.950	−1.86 ± 7.52	*p* = 0.460
	Circle 9–12	−4.36 ± 2.96	−4.85 ± 4.31	*p* = 0.864	−7.01 ± 5.86	*p* = 0.038 *
Average CMT	Average thickness	265.73 ± 8.36	263.74 ± 14.40	N/A	261.08 ± 10.12	N/A
	Circle 0–3	56.34 ± 12.69	53.41 ± 14.03	*p* = 0.854	60.21 ± 15.11	*p* = 0.783
	Circle 3–6	33.08 ± 11.24	27.02 ± 15.81	*p* = 0.420	27.31 ± 14.63	*p* = 0.601
	Circle 6–9	3.61 ± 12.80	2.01 ± 11.08	*p* = 0.813	3.01 ± 17.32	*p* = 0.882
	Circle 9–12	−9.66 ± 10.20	−12.05 ± 15.08	*p* = 0.082	−19.33 ± 20.52	*p* = 0.002 *

Values are shown as means ± SDs. DR, diabetic retinopathy; SCP, superficial capillary plexus; DCP, deep capillary plexus; CMT, central macular thickness; N/A, Not Applicable. * indicates a statistically significant difference (*p* < 0.05).

## Data Availability

The data and materials in this study are available from the corresponding author on reasonable request.

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
