# Peer review of "Retinal Microvascular Signs in Pre- and Early-Stage Diabetic Retinopathy Detected Using Wide-Field Swept-Source Optical Coherence Tomographic Angiography"

_jcm, 2022, doi:10.3390/jcm11154332_

Round 1
Reviewer 1 Report
The authors present here a study on pre-diabetic and early-diabetic patients and their results clearly reflect the benefits of using advanced OCT-A imaging. The article seems well organized and concisely presented. I would like to congratulate the authors for doing such comprehensive measurements, though it needs to be seen if similar benefits arise if the same parameters are applied to a larger and more diverse cohort of patients.
Author Response
The authors present here a study on pre-diabetic and early-diabetic patients and their results clearly reflect the benefits of using advanced OCT-A imaging. The article seems well organized and concisely presented. I would like to congratulate the authors for doing such comprehensive measurements, though it needs to be seen if similar benefits arise if the same parameters are applied to a larger and more diverse cohort of patients.
Response: We are extremely grateful to the Reviewer for these pieces of advice. Although we included about 40 eyes in each group, the number of patients is indeed small, and we plan to add severe non-proliferative diabetic retinopathy (NPDR) in the future.
Many thanks for your important and helpful suggestions on our manuscript entitled “Microvascular retinal changes in pre-and early-stage diabetic retinopathy detected by wide-field swept-source optical coherence tomographic angiography”. We hope that we have adequately addressed your suggestions and that our manuscript is now suitable for publication. Please let us know if you have any further questions or suggestions.

Reviewer 2 Report
In this cross-sectional study conducted on a small type 2 diabetic population, the authors observed that the novel and wide-field OCTA could potentially help to detect earlier diabetic microvascular abnormalities.
The manuscript is interesting. The conclusions are supported by the results. The limitations of the study are addressed by the authors in the text.
This reviewer raises a few issues to address.
1- Actually, the diagnosis and staging of diabetic retinopathy encounters several difficulties. Therefore, particularly in these times of the COVID-19 pandemic, the possibility of diagnosing DR and doing follow-up with telemedicine (1- Diabetes Metab Res Rev. 2019 Mar; 35 (3): e3113. doi: 10.1002/dmrr.3113. 2- J Diabetes Res. 2020 Oct 14; 2020: 9036847. doi: 10.1155/2020/9036847.) is an important resource, especially in geographic areas where the movement of patients to specialized centers can be long and demanding. Certainly, this issue as well as above references deserve a comment in the text.
2- The diabetic population studied, particularly with DR, has an excellent eGFR, on average greater than 100 ml/min/1.73 m2. This data is surprising considering that it is a type 2 diabetic population on average well over 50 years of age. In fact, generally the expressions of diabetic microangiopathy (retinopathy and nephropathy) are correlated. How do the authors interpret these data? Is the known duration of diabetes short? This information is neither provided in the table nor in the text. Authors should address this problem.
Author Response
The manuscript is interesting. The conclusions are supported by the results. The limitations of the study are addressed by the authors in the text.
This reviewer raises a few issues to address.
1- Actually, the diagnosis and staging of diabetic retinopathy encounters several difficulties. Therefore, particularly in these times of the COVID-19 pandemic, the possibility of diagnosing DR and doing follow-up with telemedicine (1- Diabetes Metab Res Rev. 2019 Mar; 35 (3): e3113. doi: 10.1002/dmrr.3113. 2- J Diabetes Res. 2020 Oct 14; 2020: 9036847. doi: 10.1155/2020/9036847.) is an important resource, especially in geographic areas where the movement of patients to specialized centers can be long and demanding. Certainly, this issue as well as above references deserve a comment in the text.
Response: We are extremely grateful for your expert advice. Indeed, the diagnosis and staging of diabetic retinopathy encounters several difficulties especially in the COVID-19 pandemic. However, Wide-area OCTA is a relatively large and not easy to move device. The most commonly used device for telemedicine screening DR is still fundus color photography. With the development of optical coherence tomography technology, the miniaturization of equipment and the improvement of information acquisition speed, OCTA, as a more sensitive screening method, will have broad prospects in the field of telemedicine in the future.
Changes in the text (Lines 82-85, Page 2): In addition, with the development of technology, the miniaturization of equipment and the improvement of information acquisition speed, OCTA, as a more sensitive screening method, will have broad prospects in the field of telemedicine. (1- Diabetes Metab Res Rev. 2019 Mar; 35 (3): e3113. doi: 10.1002/dmrr.3113. 2- J Diabetes Res. 2020 Oct 14; 2020: 9036847. doi: 10.1155/2020/9036847.) Relevant references have been quoted in the manuscript (clean version).
2- The diabetic population studied, particularly with DR, has an excellent eGFR, on average greater than 100 ml/min/1.73 m2. This data is surprising considering that it is a type 2 diabetic population on average well over 50 years of age. In fact, generally the expressions of diabetic microangiopathy (retinopathy and nephropathy) are correlated. How do the authors interpret these data? Is the known duration of diabetes short? This information is neither provided in the table nor in the text. Authors should address this problem:
Response: We are extremely grateful for this piece of advice and we concur with the Reviewer. We provided information of the duration of DM (months) in table 1, the average duration of the disease is about 1-2 years (Pre-DR: 8.14 ± 5.65 months; DR: 26.02 ± 12.34 months). Indeed, the data were beyond comprehension considering that the eGFR was greater than 100 ml/min in a type 2 diabetic population over 50 years of age. We examined the raw data and the medical records in detail, and found outliers that were causing things that were inconsistent with the reality of the condition. However, the retinal parameters of patients with outlier eGFR are reasonable. Therefore, in order not to reduce the sample size and in consideration of the short duration of DM and no obvious damage to renal function, WE deleted the eGFR.
Thanks again for your expert advices and helpful suggestions on our manuscript entitled “Microvascular retinal changes in pre-and early-stage diabetic retinopathy detected by wide-field swept-source optical coherence tomographic angiography”. Based on your suggestions, we have carefully addressed all the issues and have modified our manuscript accordingly. Our references to the line numbers refer to the marked-up copy that we have uploaded as a ‘Revised Manuscript with Tracked Changes’ file. All the changes have been accepted in the clean revised manuscript uploaded as a ‘Manuscript’ file. We hope that we have adequately addressed your suggestions and that our manuscript is now suitable for publication. Please let us know if you have any further questions or suggestions.

Reviewer 3 Report
The search using Plagiarism software revealed that the paper of Fabao Xu has a high level of plagiarism (42%).. The authors need to decrease below 20% to be accepted for publication.
The statistical analysis section needs improvement. First, the sample size is small. Second, the authors did not perform normality testing (even if they did). Thus they need to use only nonparametric tests, specifically the Mann-Whitney U test and Kruskal-Wallis.
Tables 2 and 3; the authors put P values but do not indicate to what comparisons they correspond. This needs to be clarified in the legend..
Figures: also need to be worked on. If a nonparametric test is used, the authors must replace the histograms with boxplots. It lacks legends...
Author Response
The search using Plagiarism software revealed that the paper of Fabao Xu has a high level of plagiarism (42%).. The authors need to decrease below 20% to be accepted for publication.
Response: Thank you very much for your advice. As suggested, we revised the manuscript and reduced the repetition rate to 16% . The Plagiarism Survey Report is attached.
The statistical analysis section needs improvement. First, the sample size is small. Second, the authors did not perform normality testing (even if they did). Thus they need to use only nonparametric tests, specifically the Mann-Whitney U test and Kruskal-Wallis.
Response: We are extremely grateful for this piece of advice and we concur with the Reviewer. Although we included about 40 eyes in each group, the number of patients is indeed small (20 patients in each group). In order to utilize all the data of the patients as much as possible, if both eyes of the patients met the inclusion criteria, both eyes were included in this study. For normality testing, the Kolmogorov-Smirnov test was used. All retinal and choroidal parameters in tables 2 and 3 fit Gaussian distributions, and a generalized estimating equation (GEE) statistical method was used to compare normally distributed quantitative variables.
Changes in the text: Lines 188-197, Page 6.
Tables 2 and 3; the authors put P values but do not indicate to what comparisons they correspond. This needs to be clarified in the legend.
Response: We are extremely grateful for your expert advice. P values represent the comparisons between the Pre-DR and Early-stage DR group and the normal control group.
Changes in the text: Lines 232-233, Page 8.
Figures: also need to be worked on. If a nonparametric test is used, the authors must replace the histograms with boxplots. It lacks legends...
Response: We are extremely grateful for your expert advice. We added legends of Figures 4 and 5. All retinal and choroidal parameters in tables 2 and 3 fit Gaussian distributions, and a generalized estimating equation (GEE) statistical method was used to compare normally distributed quantitative variables.
Changes in the text: Lines 240-242, Page 8; Lines 284-286, Page 10.
Many thanks for your important and helpful suggestions on our manuscript entitled “Microvascular retinal changes in pre-and early-stage diabetic retinopathy detected by wide-field swept-source optical coherence tomographic angiography”. Based on your suggestions, we have carefully addressed all the issues and have modified our manuscript accordingly. Our references to the line numbers refer to the marked-up copy that we have uploaded as a ‘Revised Manuscript with Tracked Changes’ file. All the changes have been accepted in the clean revised manuscript uploaded as a ‘Manuscript’ file. We hope that we have adequately addressed your suggestions and that our manuscript is now suitable for publication. Please let us know if you have any further questions or suggestions.

Round 2
Reviewer 3 Report
The authors have answered my comments
Author Response
Thank you for your recognition of our work